# Fixing Students' Early Math Skills is Critical for the Future of ML and all STEM

## Abstract

This position paper argues that fixing early math skills of students is critical for the future of machine learning (ML) and all science, technology, engineering or mathematics (STEM) professions. It also describes some ways ML and STEM scientists can help.

A common requirement for becoming a successful ML or STEM scientist or professional is fluency in primary and early secondary school mathematics (arithmetic and basic algebra in particular). Math learning is cumulative and thus, these early skills are critical both for learning high school or college math, physical sciences and computing courses well, as well as for learning to correctly write code for ML or scientific applications. Because of various reasons, including circumstances beyond their control, many students are not able to build a strong foundation in arithmetic in elementary school. This leads them to struggle with pre-algebra skills and then, with (scalar) algebra and everything after that. What can ML and STEM scientists and educators do to remove or to reduce the effect of some of the learning barriers? We describe two partial solutions. The first is running, or participating in, out-of-school math support programs (here "support" can mean tutoring sessions, math practice sessions, or just sharing free or low-cost math learning resources and encouraging practice) to help close the early learning gap between those with such math awareness and resources at home and those without. The second is that ML or STEM educators and researchers could and should step in and comment on the long-term impact of modern elementary and middle school math education policies (such as very little math practice, almost no homework, either no testing or not informing students and parents about testing, etc). Since it is practically difficult to conduct rigorous long-term education research, most current policies are based on short-term (2-3 years or less) research.

## 1 Introduction

**This position paper argues that fixing early math skills of students is critical for the future of machine learning (ML) and all science, technology, engineering or mathematics (STEM) professions. It also describes some ways ML and STEM scientists can help.**

Mathematics is a foundational skill for academic and career success, and particularly for machine learning (ML) and math-intensive STEM fields such as Engineering, Physical Sciences, and Computing [1]. Since math learning is cumulative, early gaps in math achievement tend to grow over time, leaving many students unprepared for advanced high school math courses which are, in turn, necessary for college STEM success. This is corroborated by research [1–3]. As can be seen from Fig. 1 which is taken from [2,3], students who were "on track" in 4th grade have an 82% chance of meeting 8th grade math expectations, while the ones "far off-track" in 4th grade have only a 10% chance.

Submitted to 39th Conference on Neural Information Processing Systems (NeurIPS 2025). Do not distribute.

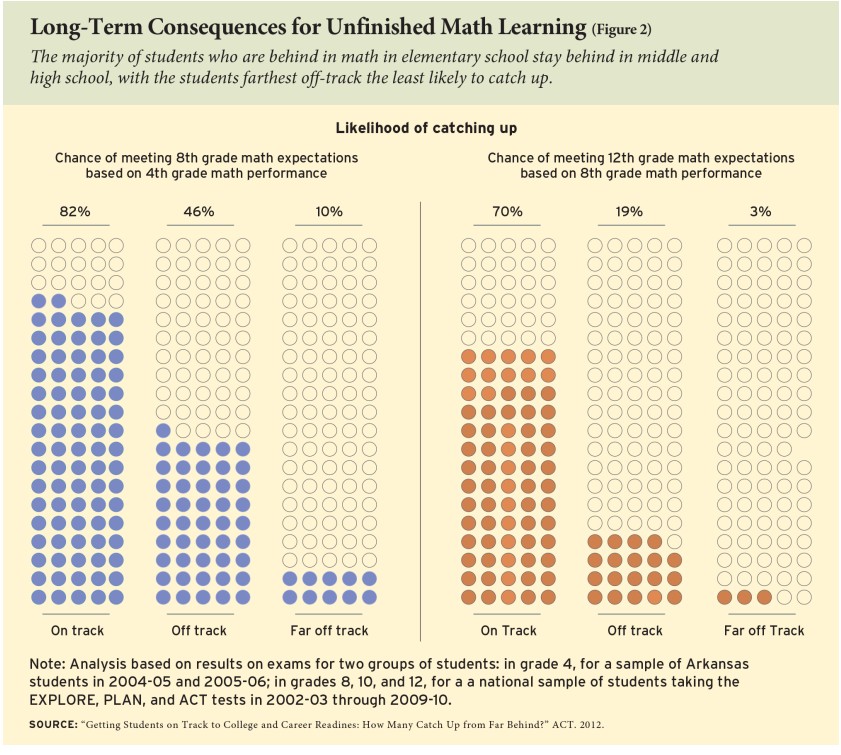

Figure 1: *Research showing that math learning is cumulative. Figure taken from [3]; it uses data from [2].*

From 8th to 12th grade, this chance reduces to 3%. We also explain this issue with a concrete example from ML below. For various reasons, including many circumstances beyond their control (e.g., family income or education levels, inadequate resources for public education, non-uniform teacher quality, COVID-19 learning losses, and an overall atmosphere of expecting less[1]), certain students face greater challenges in developing good early math skills [4, 5]. Without intervention, these math skills gaps can persist or grow, limiting their opportunities for higher education and well-paying careers. Not surprisingly, a study conducted by the Urban Institute [6] shows that improving math scores by 0.5 standard deviations for children up to age 12 is associated with larger increases in age-30 earnings than similar improvements in reading scores, physical health, or social and emotional health. For a similar reason, access to Algebra has been referred to as one of the greatest civil rights issues of our time [7].

**Why early math skills matters: an example from ML.** Understanding modern machine learning – as well as many other STEM fields – requires a good grasp of linear algebra (vector and matrix algebra), and probability and statistics, and a good ability to code algorithms based on these math concepts in MATLAB or Python. These is essential to understand, develop, and evaluate novel applications as well even when using ML toolboxes. With tool such as ChatGPT or Gemini providing the ability to automatically generate code snippets, math skills are becoming even more important. ML and other scientists need the ability to check the correctness of auto-generated code and understand which settings it may fail in. As an example, the least squares estimation problem – recover an unknown vector $\theta$ by minimizing $\|\mathbf{y} - \mathbf{X}\theta\|_2^2$, when $\mathbf{X}$ is a tall matrix – is a problem that occurs in a large number of ML and STEM applications. For example, it is used for parameter learning in linear regression as well as a sub-routine in various algorithms used in airplane navigation. Least squares estimation has a closed form solution $\mathbf{X}^\dagger \mathbf{y} := (\mathbf{X}^\top \mathbf{X})^{-1} \mathbf{X}^\top \mathbf{y}$ and code for it is two lines in Python. The code will work perfectly if the matrix $\mathbf{X}$ is well-conditioned, but does not provide a warning when it is not. Knowing that the condition number of $\mathbf{X}$ governs solution accuracy is important to realize that the code may give a very inaccurate solution when $\mathbf{X}$ is badly conditioned. Regularization techniques, such as Lasso or Ridge Regression, exist to deal with this ill-conditioning and Python

---

[1]For instance, deemphasising sufficient math practice in elementary schools, or not informing students or families that standardized test scores in fifth and later grades decide which math track a student gets placed into.

toolboxes exist to call these routines. However, realizing that ill-conditioning is a problem and which tools to use to address it requires knowing linear algebra and some statistical ML ideas.

However, one cannot learn linear algebra, or even learn to code it in, without a strong knowledge of (scalar) algebra that is typically taught in middle and early high school. Algebra also forms the foundation of the ability to understand probability concepts such as the probability mass or density function, or the ability to understand how to manipulate signals or images which are functions of time or space. Algebra cannot be understood without fluency in elementary school arithmetic (add, subtract, multiply divide; negative numbers; fractions, decimals, and four operations with these). As an example, the ability to solve for $x, y$ from two equations in $x, y$: $0.2(3x + 5) - 9x + 22 = 1/2$ and $x - y = 0.25$ requires arithmetic fluency. Solving for two variables from two linear equations forms the basis of understanding the more general setting of solving a linear equation in $n$ variables, using vectors and matrices, or to code these in.

## 2    Some Reasons for Unequal Early Math Skills

For various reasons, including many circumstances beyond their control, some students face greater challenges than others in developing good early math skills. Some examples of such circumstances include inadequate resources for (nearly) free good quality public education in most of the world, non-uniform teacher quality, family income, family education levels, COVID-19 learning losses, unconscious bias, language barriers, and a culture of low expectations [4].

Some of these barriers are beyond the control of educators, and are a public policy challenge, but the last one – culture of low expectations – is due to various design changes that were introduced to improve the modern K-8 (elementary and middle school) experience of students. As we explain later, this can be changed at least partially with simple and almost zero-cost modifications. As college STEM educators, we see that certain skills are essential for college success in math or coding heavy courses: (i) sufficient math practice done individually; (ii) completing assignments and homeworks; and (iii) studying for tests. Our interpretation of "culture of low expectations" is the following:

1. insufficient focus on math practice in elementary school during school hours;

2. almost no homework even in fifth grade or middle school;

3. not informing young students or parents about testing and not encouraging them to study for a test;

4. little information provided to families that a good grasp of elementary school arithmetic is essential for later success, and how to use free online resources to help students get this grasp (if school teaching does not suffice),

5. information on the fact that students are often tested in fourth or fifth grade in order to allow them to join advanced math tracks starting in middle school in many countries.

6. Moreover in some districts this track has been removed and this can be even worse for the high-performing students in these schools. This track, that allows students to take algebra by 7th or 8th grade, is an unsaid requirement for students to be able to do the advanced math courses in high school that are needed for success in SP, ML and, in fact all science, technology, engineering and math (STEM) fields. Access to algebra has been validly referred to as the greatest civil rights issue of our time as it allows students access to economic ladders of opportunities [7]. The last issue may be US-specific. However, a lot of the world often emulates the US education system, and a lot of education research comes out of the US, and hence its impact will be felt around the world.

## 3    Article Organization: What can ML educators and researchers do?

In the next two sections below, we discuss two types of approaches that may reduce the impact of some of these barriers.

1. One is via *out-of-school programs* that SP majors (undergraduate or graduate students, faculty, or industry professionals), and anyone with college-level math training and passion for math, can start or participate in. University support can make it easier to start such a program. We describe two such programs in Sec. 4 next.

2. However, no out-of-school program can achieve the broad reach and accessibility of public schools in serving all students. In Sec 5, we share *our suggestions for public schools*. As signal processing and engineering educators, who have been running school math support programs, these suggestions are based on what we see as the long-term impact of elementary school math teaching policies.

We describe the above two partial solutions in Sec. 4 and 5. Finally in Sec. 6, we discuss alternative viewpoints and argue why these are not always useful in the long term.

# 4 Partial Solution I: Out-of-school Math Support Programs

We describe three out-of-school university STEM faculty run, or STEM-professionals run, math programs here – CyMath, Algebra by 7th Grade (Ab7G), and Math Motivators. All information provided below is based on the programs' websites or interviews with the programs' directors and coordinators.

CyMath is Iowa State University's math tutoring and support program that started in Fall 2020. This primarily relies on volunteer graduate students or faculty as the tutors. It is a small program that provides intensive (once or twice-a-week in-person or hybrid) math support along with at-home math resources. Ab7G is an elementary and middle school math mentoring program that was started at Purdue University in 2017. Ab7G provides monthly math mentoring followed by STEM exposure labs. Most of its mentors are paid undergraduate students. Both programs provide free regular math support, along with at-home learning resources, to students in grades 3-8 while prioritizing those from difficult backgrounds. Math Motivators, run by the Actuarial Foundation, has similar goals to CyMath and Ab7G, but a slightly different approach and runs in-school tutoring sessions.

Our goal in describing these is to invite readers from around the world to start, participate in, and write about math support programs for their local schools. The authors would also like to hear from others running or interested in starting such programs so everyone can learn from each other. Our hope is also that this article encourages universities, colleges, and or professional organizations, such as IEEE or ACM or NeurIPS Foundation, to support their faculty or their membership interested in starting such programs. Also, such programs could also be started by community organizations that reach out to the community and to nearby colleges and high schools for tutors.

## 4.1 Tutoring as a useful intervention

Many studies, and meta-analyses, demonstrate that the impact of good quality math tutoring and/or mentoring can be impressive [8–15]. The work of Robinson et al [14] described a large and well-known study on math tutoring. It documented the benefits of high-dosage and well-planned tutoring provided by well-trained tutors who can bond well with students. Their work, based on pre-COVID data, describes high-dosage as three or more in-person sessions per week. While such a program was shown to be very effective, it is not scalable, and limits the number of students it benefits. Moreover, its long-term benefits were not studied. Extensive but low-dosage math mentoring, as done by Ab7G, reaches many more students. This model may not be effective for all students that participate. However, based on results from a small study [15] (described below), it could significantly benefit many of its participating students. While good quality tutoring has been extensively used and studied, many tutoring interventions end once the research study is over. The programs described below are designed to be intentionally simple and low-cost making them sustainable, scalable, and easily replicable elsewhere.

## 4.2 Algebra by 7th Grade (Ab7G) at Purdue University

Ab7G is a K-8 math mentoring program that was established in 2017 at Purdue with the goal of increasing the number of 7th grade students that are academically prepared to take algebra. It is provided at no cost to the participants. Ab7G is built upon three foundational pillars: student self-efficacy, mentorship, and parental engagement. Students enhance their mathematical proficiency through both online and in-person instruction, through resources and incentives provided to encourage practice at home, and through collaborative Engineering and STEM exposure activities (Fun-Lab). Parent workshops run in parallel and focus on topics such as exploring mathematical principles, navigating the online math tool ALEKS [16], and creating growth mindsets.

In 2017, a partnership was created with Lafayette School Corporation (LSC) to help recruit students for the program. 44 students attended in-person sessions twice per month for a total of sixteen sessions during the 2017-2018 academic calendar. The exigencies of the COVID-19 pandemic forced the program to move to a virtual platform. Ab7G now has partnerships with two other large school districts as well. Currently, Ab7G operates in a hybrid format, offering two in-person sessions and four virtual sessions each semester, all on Saturday mornings. At the conclusion of each academic year, Ab7G provides the district office with comprehensive reports on students' final performance and attendance for the sessions. The schools share participants' standardized and state-wide test scores data.

The Saturday sessions are structured for students to work on McGraw Hill's ALEKS (Assessment and Learning in Knowledge Spaces) adaptive mathematics learning software [16], mentorship with an undergraduate student mentor, in-person sessions lunch, and an Engineering exposure Fun-Lab. Ab7G participants are encouraged to work at least fifteen minutes per day most days a week on ALEKS [16] outside of the Saturday program and attempt to master three topics per week. At the beginning of each session, students are recognized for reaching their time and topic goals and honorable mentions are given to students that come close. The following are some lessons learned over the seven years of Ab7G: (1) establishing strong partnerships with schools is crucial for the recruitment and retention of participants; (2) engaging the parents can be hugely beneficial for the students; (3) providing translations for documents is useful; and (4) designing the program in a way that encourages the retention of mentors each year is essential for continuity.

**Impact.** When Ab7G started, i.e., during the 2017-2018 academic year, 44 students attended in-person sessions twice per month for a total of sixteen sessions. By 2024, Ab7G had 208 students participating. A study in [15] on Ab7G participants over the first year of the program reported very similar results. It noted significant growth in mathematics proficiency among those who completed a full grade level in ALEKS each year, n=20. It also showed (see Table 3 of [15]) that at least 5 students jumped from being at 8-36th percentile, in the standardized tests given at their school, to a 75-99-th percentile over the course of an year. Those already at a high percentile stayed in that level. This growth also correlated with achieving proficiency or higher on Indiana's Learning Evaluation and Assessment Readiness Network (ILEARN) test which is administered yearly to Indiana grades 3-8 students.

## 4.3 CyMath at Iowa State University

CyMath was founded in Fall 2020 to help support math learning of school students during COVID-19. Its long-term goal is to increase the number of youth from low-income,or other difficult, backgrounds who are prepared to pursue, and succeed in, STEM majors in college. It ran as a virtual program until 2022, when interest in online-only tutoring fell. In 2023 Fall, with help and advice from Ab7G, it was re-started CyMath as a primarily in-person program for one school in the local school district. CyMath begins tutoring students in grades 3-5 and follows them through the school years, allowing tutors to fill in the learning gaps while they are still small. The program hopes to continue supporting students through early high school at least. A significant fraction of CyMath tutors are Computer Science, Engineering, Math, or other STEM graduate student volunteers, and a smaller fraction is STEM undergraduate students (most undergraduates are paid tutors). Since 2024, these tutors are supported and trained by paid pre-service teachers (Education undergraduate students in their last two years).

Starting in 2025, CyMath runs after-school on two weekdays in one school, and it runs on Saturday mornings on either ISU campus or via Zoom.

CyMath buys accounts for McGraw Hill's adaptive math learning software ALEKS [16]. Tutors use this as a base for tutoring and for encouraging at-home math practice which is critical for math fluency and success. CyMath does not insist on a particular tutoring curriculum. Tutors are encouraged to start by asking a new student to tell them or teach them what they recently learned in school and take it from there. Grade-based standards are shared and some training on teaching math in a fun way, while encouraging a growth mindset, is provided. Students are encouraged to regularly work on math on ALEKS at home. For students who do not have reliable internet access, workbooks or worksheets are also sent home. Regular text messages and emails, along with periodic phone, or in-person chats, are used to motivate parents to support their students.Finally, when possible, CyMath attempts to inform parents and the students about the dates for standardized testing and also what adaptive testing

means (need to answer the early problems carefully to get the harder problems which are also worth more points). Lastly, CyMath runs STEM or Engineering exposure sessions once or twice a semester.

Many of the CyMath tutors are international students or immigrant faculty, including the program leads. Since the language of math is universal, these tutors have been as effective as domestic graduate students. They also bring an international perspective to math learning and expose students to world cultures and math teaching approaches, that is often completely missing in elementary schools. At the same time, the tutoring helps the tutors become better college educators (teaching assistants or instructors), provides them an understanding of the modern US math education system, and helps them find community and mentors which often positively impacts improves their wellbeing. It also enables them to pace their own college classes better.

MI-STEM graduate student or faculty volunteers – who know their math well and have a passion for it – are an untapped high-quality volunteer tutoring resource since they are already funded as research or teaching assistants. They are more mature, likely to continue for longer, and can support various math skill and grade levels. Also, the mentoring is informed by the math skills needed for success in ML and STEM course-work, and the students also learn about STEM research careers from their mentors.

**Impact: results of preliminary evaluation of the program provided to us by its director.** CyMath welcomes all students, but also prioritizes low-income students or those from other difficult backgrounds.

For the first online-only iteration of CyMath, a total of 14 Engineering or Math graduate student volunteer tutors tutored about 35 children for some part of the two years from 2020-22. In Fall 2023, the in-person, and later hybrid, CyMath started for one school in the local school district. As of November 2024, this hybrid-mode CyMath had 20 students – of these 9 were Black, 4 were Latino, 4 were Asian, and 3 were White. All received at least once a week of in-person tutoring. Out of 20 students, at least 8 had parents who speak two or more languages, and 2 of these eight themselves were not fluent in English yet. The tutor group was equally diverse and consisted of three Black, two Latino, five White, and many Asian tutors; at least one Arabic speaking tutor, and at least three Spanish speakers. The tutors grew up in four different continents – North America (US), Latin America (Guatemala, Ecuador), Africa (Egypt, Uganda, Sudan), and Asia (China, Korea, Taiwan, Thailand, India, Nepal, Iran).

The program compared pre and post test scores on standardized tests administered by the school district. For the 2023-24 year, these were MAP (Measures of Academic Progress) tests while in the next year, the testing switched to iReady. The program has seen encouraging success for some of its students. *Of the six students who started with CyMath in Fall 2023, one has transitioned from being at a 40-th percentile in September 2023 to being at the 85th percentile in May 2024 and at the 92-nd percentile in January 2025. A second student from this cohort has been showing more modest but steady gains and has gone from 20-th to 38-th to now 60-th percentile.* One newer student who started in Fall 2024 has also gone up from 34-th to 66-th percentile, while some of the other newer students are going up by 5-10 percentile points. However, since percentile measures are noisy (the score gaps are very small), only large and consistent changes are meaningful.

During Spring 2025, CyMath has enrolled many more middle school students for a total of about 40 students in grades 3-8.

### 4.4 Math Motivators: run by the Actuarial Foundation

Math Motivators is a math tutoring program run by the Actuarial Foundation and provides in and out of school math tutoring support to students from disadvantaged backgrounds in various schools around the US. It is supported by the Society of Actuaries (a world-wide organization of actuaries), the American Academy of Actuaries, and various corporate sponsors. Approximately 75% of its participating students ares eligible for free or reduced school lunch, and 70% identifying as Black, African American, Latino or of Spanish origin. Math Motivators offers free in-person math tutoring to under-served students in grades 3-12 who need and want tutoring but otherwise cannot afford it. Two students of similar ability are paired with one tutor, who works with the students to help them become proficient in math. Some of their programs offer tutoring during the school day while others offer tutoring after-school. MM currently runs in eight cities in the USA.

### 4.5 Conclusion for this part

In conclusion, the best that one can expect from out of school support programs is large gains for some students and small gains for most students.

It would be interesting to study the impact if many STEM groups such as the NeurIPS Foundation, the International Machine Learning Society (IMLS), IEEE and/or ACM partner with one of these programs, or start their own math support program. These professional groups together likely have a much larger membership base than that of the actuarial societies.

## 5 Partial Solution II: Comment on Long-Term Impact of School Math Education Research and Policies

Out-of-school programs cannot reach all the students that public schools reach. Even if there were infinite resources available to teach and transport all children, many will not want to, or will not be able to, join one due to other commitments. Hence the way to impact the math learning, and future success, of all students and especially those from the most difficult personal circumstances, is to work to help change school policies around us. While school math curriculums differ significantly across the world, and even within the same country, the math background needed to succeed in college ML or STEM courses is much more similar around the world. Furthermore the same is true for the skills needed to succeed as an ML or STEM professional worldwide.

- *Math Homework and Sufficient In-School Math Practice.* Individual math practice is essential for math fluency. Thus, sending reasonable amounts of math practice work home as homework can have significant benefits, especially when teachers have the time to check the students' work carefully and discuss it with them so that student learning can be corrected early and easily. College courses, and often high school courses, do have significant homework and this would also teach students an excellent skill needed for college success. The literature on elementary school homework is mixed. Some studies, such as [17] (based on Latin American schools) or [18] (based on China's schools) argue that reasonable amounts of math homework in primary school does, in fact, improve student achievement. The latter also shows that too much homework does not provide extra benefit. On the other end, studies such as [19] conclude that homework does not help in elementary school. However, as we argue in the section on Alternative Views below, Sec 6, this and other similar studies are not long-term studies.

- *Parent and Student Awareness – Early Math Skills, Testing.* Many parents are not aware that a good early foundation in math is important, or how to help their child build such a foundation. This knowledge should not be the privilege of children or relatives of STEM professionals. It should be provided to everyone by the schools. Moreover, it will automatically get conveyed if math is emphasised, homework is sent, and if free workbooks, printed worksheets, and information about good online resources such as Khan academy or k5learning, are shared with families. Some information is summarized here <add web ref>

- *Testing in Schools and Informing Students and Parents.* In many elementary and middle schools, there are few tests other than the standardized tests. Parents and students are often not informed about the standardized test dates, and are not encouraged to study for the tests. This is done even though the same tests' results are used later to place students into different math tracks. However, if a parent asks, they are provided all of this information. Consequently, we end up disadvantaging a much larger number of math-capable students who may have otherwise come better prepared for these tests. Also, good test taking skills, including studying for the test, are essential for college success. An impact of these policies is that many math-capable students get shut out from accelerated math classes which are often considered essential for college STEM programs.
  There are good intentions also for the above policies, we explain and discuss these below in Sec. 6, where we again argue that in the long-term, these policies likely end up having the opposite effect.

- *Use of summer to make-up for lost learning.* Despite all efforts, some students may require additional support to stay on track. The summer, and especially the summer before middle school, can be used efficiently to help students catch up on arithmetic and pre-algebra

skills for example. Summer is also the time many undergraduate and graduate students and faculty are more available and schools can tap into these resources by partnering with nearby colleges or universities.

All of the above policies were designed with good intentions and we describe the intents below in the section on alternative viewpoints.

# 6 Alternative Views

Most of what we discuss here are authors' viewpoints. We can remove or shorten this section significantly based on reviewer comments.

All modern elementary and middle school education policies are based on quantitative research, i.e., research that tests the validity of a hypothesis or the usefulness of a new approach by collecting data (student grades, survey data from students, teachers, parents) on its benefit. Thus, on first glance, these policies should be a good idea. However there are a few practical limitations of most of these studies.

1. Most elementary education studies are not long-term studies, e.g., [15, 19, 20]. The reason there are no long term, say 10 year studies, and not many even 5-year studies either, is that it is all but impossible to conduct such studies in a statistically sound fashion. School systems and hence student data in many countries such as the US are decentralized. Students move to different states often and different schools' data is often not nationally shared to protect student privacy.

   A related point is that elementary education, secondary math education, and college math and STEM education are often treated as entirely different research and teaching areas. Elementary education researchers do not even study the impact of their proposed ideas on sixth grade math outcomes. Middle school math teachers do not usually comment on elementary school math policies. This is the case for example for the studies cited above.

2. Research studies focus on improving one aspect of learning but often disregard other aspects such as the financial cost, the time-cost, the motivation levels and the attention spans of young students. As an example, a study focused on improving student engagement or making the school experience more fun disregards other aspects. For instance, it ignores the fact that some schools will have much fewer resources or that after the research study is over, a lot of the extra resources such as Education graduate students will disappear. Or the fact that teachers have limited time to plan things and time that could have been spent on encouraging math practice or checking students' work, instead goes towards making the students' day fun.

   The above would not be an issue if only the best research ideas (and those with the least likelihood of harming long-term learning outcomes) were adopted in practice. In Education literature, research often gets adopted by for-profit curriculum companies who often work on convincing school districts to adopt their tools and use their recommended policies when using their testing tools.

## 6.1 Almost no homework in early grades

Studies such as [19] conclude that homework does not help in elementary school, while others such as [20] argue that it is useful only if it is high quality. None of studies suggest that there is any negative impact of giving homework. These studies followed students for a short period of time (less than two years) and do not tell us what the impact of learning to do homework is on college or high-school success. This is the case also for a lot of elementary education literature, since it is not easy to do long-term, e.g., 10 year, studies for reasons explained above. For example, homework given in fourth grade may or may not improve math achievement by much in fourth or fifth grade. However, maybe it teaches good study skills that are beneficial later in high school or college.

Another argument against homework is that it is unfair to expect homework getting done when some children will get parental help, while others will not. However, this issue can be partly addressed by evaluating and grading students only on in-class testing and not on homework. Moreover, a longitudinal study [21] argues that the impact of parental help is limited and homework is in fact

beneficial to all students' learning. Another issue, that research studies do not account for, is the fact that many parents (with awareness and resources) make their child practice math at home even without school homework or that much of the world schools do still make students do homework. Thus, even though the goal of education research is to improve outcomes for all students, and especially those with less, in practice, when looking at the long-term impact, these end up creating unequal outcomes for the students it seeks to help.

A third argument against homework is that some students may not have quiet space at home to complete it. This is true. However, this would remain true even in high school.

## 6.2 Little math practice and only math fun in early grades

Most of what we discuss here are authors' viewpoints. Besides homework, providing enough in-school individual math practice can also help with student learning in younger grades. This would be easier for teachers to incorporate if they were not encouraged convert every math lesson into a game. Planning for fun math games takes a lot of teacher time and uses up a lot of classroom time. It likely takes away the class time that could otherwise be spent by the students on math practice; and by the teachers on checking the students' work and correcting errors early.

There is a line of literature that argues for making math fun using games in order to engage little students. This is surely beneficial in the short term and is a great idea if there was enough time and resources. However, in practice this comes at the cost of other things in a strained public school budget system. Often it means that there is little time left for having students do sufficient practice in simple concepts. Practice and individual practice builds fluency which is needed for understanding algebra well. As argued earlier, algebra is the basis of all STEM and all ML. There are no studies that evaluate the long-term (say 10 year) benefit of play-based math versus the old-fashioned math practice drills for reasons explained below.

Making math only fun in early grades means students start expecting the same in middle and high school, and even college. This is another reason why many students struggle with algebra which is the first abstract concept they see. There are ways to improve engagement in algebra too, e.g., using coding tools. However, this again comes at a cost (finite school time and budget, even learning to code requires learning other tools). One can go to block-based coding tools to make things even more fun. However in the process there is no time left to teach enough.

## 6.3 Not informing parents and students of testing

The policies to not bother parents were also created with good intentions to keep things fair and equal for all students since some may not have quiet study spaces at home. However this issue does not go away in college. These also have research to back them up, however, the research once again did not (and cannot easily practically) look at the long-term impact. By not informing parents and students, the schools also end up disadvantaging many students and families who could find a way to study (e.g., use a public library) if they were informed about the need to.

A second reason given to not inform students or parents about testing is that schools do not want students to only study to the test. It would be indeed be ideal if every student studied for the love of learning. However, this is rare. Most young students in most of the world study to get good grades. We believe this is still better than students never studying which is what the current system encourages. Having a weak background in early math coupled with not so good study skills taught in early school grades, result in students struggling with advanced concepts in high school and college courses.

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
