# OpenReview forum: "Fixing Students’ Early Math Skills is Critical for the Future of ML and all STEM"
_NeurIPS.cc/2025/Position_Paper_Track — Submitted to NeurIPS 2025 Position Paper Track_

### Official Review · Reviewer_jYcB · 2025-07-30

**Significance:** 3
**Presentation:** 2
**Rating:** 6
**Confidence:** 4

**Summary:**

This paper argues that students’ facility with arithmetic and early algebra is a bottleneck for the entire machine‑learning talent pipeline. Drawing on longitudinal data that show only 3% of 4th‑grade pupils who start “far off track” catch up by 12th grade, the authors frame basic numeracy as both an equity issue and an economic imperative for the ML community. They outline two avenues for action. The first is direct participation in scalable, low-cost tutoring initiatives such as Purdue’s Algebra by 7th Grade (Ab7G), Iowa State’s CyMath, and the Actuarial Foundation’s Math Motivators, whose early cohorts reportedly moved individual students from the 8-36th to the 75-99th percentile on standardized tests. The second is policy advocacy: encouraging reasonable homework, transparent testing, and summer catch-up programs, while challenging “no homework” or “all‑game” paradigms that may look beneficial in short two-year studies but undercut long-term readiness for college-level math and coding requirements.

**Strengths:**

1. It backs its argument with three hands, on tutoring programs (Ab7G, CyMath, and Math Motivators) and reports percentile jumps that make the problem and the fix feel tangible rather than abstract.
2. The authors do not just diagnose the issue; they outline low‑cost, volunteer‑friendly playbooks that graduate students or professional societies could start using next semester.
3. By focusing on students who lack math capital at home, the paper frames early numeracy as both a civil‑rights concern and a workforce imperative.

**Weaknesses:**

1. Most evidence comes from small, US‑based cohorts.
2. Cost, teacher workload, and political feasibility get only cursory treatment, leaving readers to guess how the proposals would fare in budgets already under strain.
3. Schools from other countries are barely mentioned, so readers from other countries may struggle to see the fit.

**Questions:**

1. Could the authors share any longitudinal tracking that links authors' tutoring students to later enrollment in advanced high‑school math or STEM majors, so that readers can gauge true downstream impact?

2. How might the authors' recommendations adapt to settings where reliable internet access, volunteer graduate students, or standardized testing regimes are absent or very different?

3. Have the authors considered integrating adaptive learning systems powered by LLMs as a cost‑effective substitute when human tutors are scarce, and if so, what safeguards would the authors propose to maintain instructional quality?

**Alternative Position:**

Yes, and alternative positions are well-considered and addressed by the argument

**Author Identification:**

No.

**Context:**

3

**Discussion:**

3

**Ethics:**

["NO or VERY MINOR ethics concerns only"]

**Position:**

Yes, the paper argues for or against a position related to machine learning.

**Support:**

3

**Thoroughness:**

4

---

### Official Review · Reviewer_VmzS · 2025-08-07

**Significance:** 2
**Presentation:** 2
**Rating:** 5
**Confidence:** 4

**Summary:**

This position paper asserts that strong early math skills are essential for success in college STEM courses and machine learning careers. It reviews longitudinal evidence showing that gaps in elementary math compound over time, leading to struggles in college gateway courses. The authors diagnose root causes—curriculum misalignment, lack of feedback, and inequitable instruction—and survey interventions (in-school reforms, out-of-school tutoring, adaptive online platforms), linking each to its intended impact. They call on the ML community to partner with educators and policymakers to build and evaluate scalable, data-driven math support systems as critical infrastructure for a diverse future STEM workforce.

**Strengths:**

1. Strong linkage shown between early‐grade math proficiency and later success in college‐level STEM and ML courses.

2. Identifies curriculum misalignment, insufficient feedback loops, and inequitable instruction as root causes of underperformance.

3. Surveys a broad spectrum of interventions (in‐school reforms, out‐of‐school tutoring, adaptive platforms) and directly ties each to the barrier it addresses.

**Weaknesses:**

1. Programs are described without isolating their independent impact from concurrent factors (e.g., family support, school context).

2. Little discussion of how student aptitude, motivation, or background modulate intervention effectiveness.

3. Lacks a thorough plan for preventing digital-divide or resource-gap effects in deploying online or tech-based solutions.

**Questions:**

1. How does your framework accommodate differing innate abilities and learning rates across students?

2. Do you plan longitudinal pilots with built-in data collection to quantify each intervention’s causal effect on college STEM outcomes?

3. What strategies will ensure that interventions don’t widen gaps for students lacking reliable internet or device access?

**Alternative Position:**

Yes, and alternative positions are well-considered and addressed by the argument

**Author Identification:**

No.

**Context:**

3

**Discussion:**

3

**Ethics:**

["NO or VERY MINOR ethics concerns only"]

**Position:**

Yes, the paper argues for or against a position related to machine learning.

**Support:**

3

**Thoroughness:**

4

---

### Official Review · Reviewer_EQya · 2025-08-09

**Significance:** 3
**Presentation:** 2
**Rating:** 4
**Confidence:** 3

**Summary:**

The paper argues for more involvement by the ML community in early mathematics education. The authors discuss two possible avenues for involvement. The first is at the level of personal involvement in math activities and tutoring in elementary and middle school. The second is broader political involvement by ML practitioners and their societies to chang current practices in math education in public (and private?) schools.

**Strengths:**

The position paper addresses an important issue, math education, and it connects early childhood math ability with long-term math ability, which impacts the number of people able to pursue a career in machine learning.  In that sense, it is addressing a meta-topic in machine learning, one that could impact the field in the long-term.

The paper provides fairly clear guidance on what members the ML community could do in order to institute change in line with the position.

**Weaknesses:**

The "Alternative Views" section seems to be an expansion of section 2.  The topics there seem to be intended to support the position of the authors.  As such, it's not clear they should be labeled as 'Alternative Views'.  The alternative views should be ones that argue against the primary position of the paper.  I note that the authors include arguments from both sides of the issues highlighted in section 2 and expounded on them in section 6. But most of section 6 is about refuting the alternate views.

line 400: Some strong statements without reference to literature.  The authors might want to be more explicit that this is an opinion and not a fact supported by educational studies.  If the statements are supported by prior studies, the authors should cite those.

Section 4 spends a lot of space describing in detail what seem to be relatively small programs. The author's might be well-served to summarize the programs in section 4 more succinctly, giving more space for other material (or shortening the paper).

**Questions:**

Is this paper a call for political action by members of the ML community?  That seems like the only way to bring about change in math education.  Or is there another way to influence math education?  Is this where we need longer term longitudinal studies?

**Alternative Position:**

Yes, and alternative positions are well-considered and addressed by the argument

**Author Identification:**

No.

**Context:**

2

**Discussion:**

3

**Ethics:**

["NO or VERY MINOR ethics concerns only"]

**Position:**

Yes, the paper argues for or against a position related to machine learning.

**Support:**

2

**Thoroughness:**

4

---

### Note · Authors · 2025-08-29

**1-11 Submit Again:**

Probably yes

**1-1 Submission Process:**

4

**1-2 Next Year:**

More focused topics.  Something on
"How to raise a more diverse and better trained ML workforce -- what can we do at the school (K-12) level",
"Do early math skills matter for future ML research success"

**1-3 Future Development:**

More focused topics. Something on
"How to raise a more diverse and better trained ML workforce -- what can we do at the school (K-12) level",
"Do early math skills matter for future ML research success"

**1-4 Interest:**

["Panel discussions with other position paper authors", "Structured debates on controversial topics", "Workshops for developing position papers"]

**1-5 Thoughtful:**

8

**1-6 Supportive:**

8

**1-7 Technical Aspects Versus Position:**

8

**1-8 Gate Keeping:**

8

**3-1 Review Response1:**

EQya

**3-2 Reaction To Review1:**

"The Alternative Views section should be arguing against primary position of the paper. I note that authors include arguments from both sides of the issues highlighted in section 2 and expounded on them in section 6. But most of section 6 is about refuting the alternate views."
We have discussed alternate views but will add more based on these comments.  We point out however that, as a position paper, we do believe that there are legitimate issues with the alternate views that are prevalent, and thus chose to highlight them. However, we will edit the paper based on these comments.

"line 400: Some strong statements without reference to literature. "
We will make this clearer and we will add citations for the "making math fun" ideas from literature (Mathematical Mindsets: Unleashing Students’ Potential through Creative Math, Inspiring Messages and Innovative Teaching by Boaler is one citation). Our point is that these ideas are great if infinite resources and time were available. But not in a constrained (money and time) budget system which is what most of the world deals with.

"Section 4 spends a lot of space describing in detail what seem to be relatively small programs The author's might be well-served to summarize the programs in section 4 more succinctly, giving more space for other material (or shortening the paper)." We will definitely shorten the final paper and move extra details to an Appendix or just to an online link. This is our first time submitting a position paper to NeurIPS.

"Is this paper a call for political action by members of the ML community"   This paper is in part a call to action by the ML community. How? Signature campaigns to state or country  education boards. It would also be helpful if ML professional organizations joined hands with IEEE and ACM and together published some statements on the STEM perspective on K-12 math that are noticed by government agencies. The point is to highlight the severity of the issue at stake.

Thoughtful

**3-3 Review Response2:**

VmzS

**3-4 Reaction To Review2:**

"Programs are described without isolating their independent impact from concurrent factors (e.g., family support, school context)."
All described programs are quite small; our goal is not research. The cohorts are not large enough to isolate this impact. The reason is students from difficult backgrounds move a lot, schools are decentralized, and it is not easy to control for the effect of family incomes or education levels etc.

"Little discussion of how student aptitude, motivation or background modulate intervention effectiveness."  Our thesis is tutoring and a little mentoring to practice at home can help each student reach “their personal best” (may be different for different students). We have mentioned this but will emphasize more.

"Lacks a thorough plan for preventing digital-divide or resource-gap effects in deploying online or tech-based solutions." We have mentioned that we do supplement with workbooks (inexpensive) for those without device access. We should emphasize also that there are a very large number of students who do have internet/device access, but are still not being left behind because schools do not share anything with parents (in trying to protect parent time). Parents get no suggestions on how to support their child’s learning (e.g., use Khan academy which also has videos in many languages), even when they ask the teachers. Our programs are able to help such students significantly.

"How does your framework accommodate differing innate abilities and learning rates across students?" See above response.

"Do you plan longitudinal pilots with built-in data collection to quantify each intervention’s causal effect on college STEM outcomes?"
 Longitudinal studies would be great. But as argued in the article, we believe these are not easy to conduct due to various practical barriers.

"What strategies will ensure that interventions don’t widen gaps for students lacking reliable internet or device access?" See earlier response.

Thoughtful

**3-5 Review Response3:**

jYcB

**3-6 Reaction To Review3:**

“Most evidence from small cohorts”: We do not yet have statistically sound data to back our claims. Also, longitudinal studies in education are not practical for various reasons – students move a lot and schools are decentralized. We discuss this point in alternate views section of paper.

“Schools from other countries are barely mentioned” In fact, our thesis is that US needs to learn from other countries’ math education systems. We cite outside-US studies and make this point. We will edit our writing to add more on how our ideas are relevant worldwide - math equity is an issue outside US also. Tutoring programs can be run, and are being run, everywhere. We are just not aware of university-run large scale programs.

“Teacher workload”: Our suggestion is to have students practice more math regularly using worksheets or apps in math time, and of course get sufficient breaks and play time. It is likely to save teacher time, who currently spend a lot of time on making all math concepts "fun".

“Tutoring cost” Costs are minimal if university and school administrators are on board and they advertise the program to potential tutors and students in need respectively. If the idea has the backing of an ML professional community, or if many top universities decide to run such programs, these ideas would gain significant traction. Tutoring costs are just undergrad program assistants and tutors’ hourly pay to support the volunteer graduate/postdoc/faculty tutors.

“adapt to different settings without internet or grad students” We will address this better. We do not recommend a curriculum or a tutoring approach. All math teaching styles are good and useful. Our goal is that many org's run programs that provide partial replacement for what a parent provides.

“LLM tutors”: We use an adaptive math learning app, during tutoring and for at-home practice. This helps us not worry about non-uniform tutor quality and we can group multiple kids when needed.

Thoughtful, Supportive

---

### Meta-Review · Area_Chair_k4DS · 2025-09-16

**Rating:** 4
**Confidence:** 4

**Strengths:**

This paper argues that fixing early math skills of students is critical for the future of machine learning (ML) and all science, technology, engineering or mathematics (STEM) professions. It presents results from literature showing that math learning is cumulative, i.e., being "on track" at early stage is important for success later on, discusses reasons for unequal early math skills, and proposes two classes of solutions to address the issue: (1) out-of-school math support programs, and (2) changes in education research and policy. A few alternative view points have also been covered in the paper.

1. Overall, the position paper addresses an important issue, math education, which could impact the number of people able to pursue a career in ML in a long run.

2. The linkage between early‐grade math proficiency and later success in college‐level STEM and ML courses is in particular an important observation.

3. The paper provides clear guidance on what members of the ML community could do in order to institute change in line with the position.

**Weaknesses:**

The reviews have focused on the following weak points:

1. Insufficient discussion on alternative views - the "Alternative Views" section seems to be supporting the authors' argued position instead of against it which is what "alternative views" are intended for.

2. Effectiveness of the out-of-school math support programs are described without isolating their independent impact from concurrent factors and student motivation or a thorough plan for preventing digital-divide or resource-gap effects in deploying online or tech-based solutions.

3. Most evidence comes from small, US‑based cohorts. And only little discussions are included on the cost, teacher workload, and political feasibility of the proposed solutions.

The author survey has clarified on some of the issues (including the questions below). The discussions should be incorporated into the next version of the paper.

**Questions:**

The reviewers have raised a few questions in the detailed implementation of the proposed solutions, such as how to accommodate differing abilities and learning rates across students; how to track and evaluate the impact of the intervention programs, how to prevent from widening gaps for students lacking reliable internet access; how to accommodate different availability of volunteer graduate students or testing regimes.

Another question raised is if there was any longitudinal tracking that links authors' tutoring students to later enrolment in advanced high‑school math or STEM majors - this would help emphasize the argued position.

The authors should also clarify if they meant to call for political action by members of the ML community to bring about change in math education.

Beyond these questions raised by the reviewers, I fail to see how this paper addresses the assessment criteria that "submissions to the position paper track will be judged primarily on whether they present a compelling position that warrants greater exposure within the machine learning community". Early math education is no doubt critical, but I'm not sure if there is much in the argued position that wasn't well acknowledged by the ML community already.

**Ethics:**

None.

**Thoroughness:**

4

---

### Decision · Program_Chairs · 2025-09-26

Reject